# Coenzyme a Biochemistry: From Neurodevelopment to Neurodegeneration

**DOI:** 10.3390/brainsci11081031

**Published:** 2021-08-02

**Authors:** Luca Mignani, Barbara Gnutti, Daniela Zizioli, Dario Finazzi

**Affiliations:** 1Section of Biotechnology, Department of Molecular and Translational Medicine, University of Brescia, Viale Europa, 11, 25123 Brescia, Italy; l.mignani001@unibs.it (L.M.); b.gnutti001@unibs.it (B.G.); daniela.zizioli@unibs.it (D.Z.); 2Clinical Chemistry Laboratory, Spedali Civili di Brescia, 25123 Brescia, Italy

**Keywords:** coenzyme A, pank2, PKAN, coasy, COPAN, neurodegeneration, iron

## Abstract

Coenzyme A (CoA) is an essential cofactor in all living organisms. It is involved in a large number of biochemical processes functioning either as an activator of molecules with carbonyl groups or as a carrier of acyl moieties. Together with its thioester derivatives, it plays a central role in cell metabolism, post-translational modification, and gene expression. Furthermore, recent studies revealed a role for CoA in the redox regulation by the S-thiolation of cysteine residues in cellular proteins. The intracellular concentration and distribution in different cellular compartments of CoA and its derivatives are controlled by several extracellular stimuli such as nutrients, hormones, metabolites, and cellular stresses. Perturbations of the biosynthesis and homeostasis of CoA and/or acyl-CoA are connected with several pathological conditions, including cancer, myopathies, and cardiomyopathies. In the most recent years, defects in genes involved in CoA production and distribution have been found in patients affected by rare forms of neurodegenerative and neurodevelopmental disorders. In this review, we will summarize the most relevant aspects of CoA cellular metabolism, their role in the pathogenesis of selected neurodevelopmental and neurodegenerative disorders, and recent advancements in the search for therapeutic approaches for such diseases.

## 1. Introduction

The intracellular level of acetyl-coenzyme A (acetyl-CoA) is a key factor in the maintenance and control of cellular life, being critical to a broad range of cellular processes. It is estimated that CoA and its thioester derivatives are involved in about 4% of cellular reactions in bacteria and eukaryotes [1,2]. Acetyl-CoA is the product of multiple catabolic reactions and the precursor for many anabolic processes; thus, the acetyl-CoA/CoA ratio reflects and controls the general energetic state of the cell. At the same time, acetyl-CoA is the only donor of acetyl residues for protein acetylation and modulates fundamental pathways including mitosis, cell death, and autophagy. Acetyl-CoA is formed by an acetyl moiety (CH_3_CO) bound to CoA through a high energy, thioester bond. While the di-carbon moiety comes from different catabolic and anabolic pathways, CoA intracellular levels depend upon factors such as the rate of its biosynthesis and degradation, the transport of the molecule across membranes within cells to establish and maintain various intracellular pools, the consumption of the molecule for protein post-translational modifications.

## 2. Coa Homeostasis

The intracellular CoA concentration depends upon the rate of synthesis, degradation, distribution, and usage of the molecule. Many of these processes are well characterized, but relevant aspects in their regulation and control still wait for further understanding.

## 3. De Novo CoA Biosynthesis

The CoA molecule comes from a specific de novo biosynthetic process that is highly conserved across kingdoms (Figure 1) [3,4].

Vitamin B5 or pantothenate (Pan) is the precursor of the biosynthetic pathway. The molecule is largely diffused in biology [5] and in humans, Pan deficiency may occur only as a consequence of severe malnutrition. Bacteria, fungi, and plants can synthesize Pan de novo, while animals get the molecule from the diet, the gut flora, or by degradation of endogenous CoA [6,7,8,9]. In mammals, a unidirectional, sodium-dependent multivitamin transport mechanism mediates the intestinal absorption of the vitamin [10]. In the cells, CoA synthesis starts with the phosphorylation of Pan to 4′-phosphopantothenate (P-Pan). This first reaction represents the major rate-limiting and control step in the entire process [11,12]. P-Pan is then condensed with cysteine to form 4′-phosphopantothenoylcysteine. A decarboxylation reaction leads to 4′-phosphopantetheine (P-Pant), which is firstly adenylylated to dephospho-CoA and finally phosphorylated to CoA. Each step is catalyzed by specific enzymes: Pantothenate Kinase (PANK), Phosphopantothenoylcystene synthetase (PPCS), Phosphopantothenoylcystene Decarboxylase (PPCDC), Phosphopantetheine Adenylyltransferase (PPAT), and dephospho-CoA-kinase (DPCK). A single bi-functional enzyme, named CoA synthase (COASY) carries out the latter activities in organisms like humans, mice, zebrafish and fly [13]. The synthesis of CoA is thought to take place in the cytosol, where most of the enzymes are found [14]. Recent data show that both PANK and COASY enzymes may have other localization in some eukaryotic cells, thus raising the possibility of compartmentalized biosynthetic processes [15]. In humans, there are 4 active PANK isoforms: PANK1a and PANK1b derive from the same gene with an alternative starting exon. The first is localized mainly in the nucleus, the latter in the cytosol, and the endosomal compartment; PANK2 resides in the intermembrane space of mitochondria but can be detected also in the nucleus and in the cytosol. Mutations in this gene are linked to a rare form of neurodegeneration with brain iron accumulation. PANK3 is found exclusively in the cytosol [16,17,18]. Each PANK isoform presents a different tissue distribution in mice and humans: PANK1 is expressed at the highest level in liver and kidney, followed by heart and skeletal muscle; PANK2 and PANK3 have a ubiquitous distribution, with higher levels in the liver and brain [19]. According to the present knowledge, these enzymatic activities appear to be redundant and can compensate each other, at least partially, to maintain adequate CoA level, as suggested by the mild phenotype shown by mice carrying deletion of a single *Pank* gene [17,20,21]. At the same time, double knockout mice are embryonic lethal or die soon after birth [19,22]. Whether this holds true in humans is not known for sure. It is somewhat surprising that mutations in human *PANK2* lead to a severe form of neurodegeneration, with frequent onset in the infanthood. As for COASY, there are different isoforms generated from a single gene: COASY alpha shows ubiquitous expression, while the beta form is present mainly in the brain [23,24]. A third isoform (gamma), possessing only the DPCK domain has not been extensively characterized yet. Data about the intracellular distribution of the enzyme are controversial, supporting either anchoring to the outer mitochondrial membrane [13,24] or residency in the mitochondrial matrix [23]. At the moment, the other enzymes involved in CoA biosynthesis appear to reside exclusively in the cytosol [3], but a more thorough investigation could help at defining this important aspect of CoA production. While the above-described five-step process represents the major source of CoA in most organisms, alternate routes seem to be possible. In particular, P-Pant may represent a metabolite able to enter the cells and feed the biosynthetic pathway downstream of PANK activity. Evidence from *C. elegans*, *Drosophila Melanogaster*, and mammalian cells show that P-Pant can derive from extracellular degradation of CoA catalyzed by ectonucleotide pyrophosphatases (ENPPs) [7]; in this context, the molecule appears to be stable enough to enter the cells and rescue intracellular CoA deficiency due to enzymatic defects of the biosynthesis pathway upstream of COASY. Interestingly, the metabolite can be produced and secreted from bacteria in the gut [25], possibly representing an alternative source for a precursor of CoA production [7,26]. If the molecule can freely diffuse through the membranes, it could support the production of CoA in different compartments even in the absence of the full set of enzymes required for the de novo synthesis. Furthermore, P-Pant represents a possible treatment strategy for the neurodegeneration associated with *PANK2* mutations [27] (see below).

The regulation of CoA biosynthesis is exerted mainly at the step of Pan phosphorylation by PANKs. The level of the final products of the path, CoA and its thioester derivatives, modulates the activity of the kinases and thus connects them to the flux of the oxidative metabolism in the cell. Studies on mouse and human PANK enzymes indicate that each isoform presents a specific control by these metabolites, with an IC_50_ that appears to depend upon the NH_2_-terminal domain of the protein. While mouse PANK1 Beta is weakly inhibited by acetyl-CoA [11], the alpha form, which possesses a longer NH_2_-terminal domain, is negatively and more strongly affected by CoA, acetyl-CoA and longer thioester derivatives [28]. Acyl-CoAs are strong inhibitors of human PANK2, with an IC_50_ of about 1 μM [19]. The same is true for PANK3. Given the intracellular concentration of CoA thioesters, the enzymes should be inactive most of the time, unless other molecules release the blockage. This is the case for PANK2, which is localized in the intermembrane space of mitochondria where long-chain acylcarnitine competitively antagonizes the negative regulation of the kinase by acetyl-CoA [29]. Acyl groups transferred by the carnitine shuttle and CoA are essential substrates for mitochondrial beta-oxidation and this type of regulation connects PANK2 activity with the status of this essential catabolic pathway. In addition, the final step in CoA production represents another point of regulation of the pathway, particularly when the flux through the pathway is increased; CoA and CoA thioesters exert a feedback inhibitory control on mammalian COASY activity [15].

The knowledge about PANK enzyme’s biological role has grown significantly in the latest years, but many aspects deserve further attention and understanding. The connection between their specific and different intracellular localization with the distribution of CoA in the same compartments and the capacity of each enzyme to compensate for the absence of other isoforms represent issues that wait for a better comprehension; their description could improve the understanding of disorders linked to perturbation of CoA homeostasis.

## 4. CoA Degradation

CoA in the diet represents the main source of Pan. Its intestinal degradation starts with the dephosphorylation to dephospho-CoA mediated by an alkaline phosphatase. ENPP enzymes then hydrolyze the diphosphate bond in the molecule generating P-Pant [30]. The molecule can be absorbed or further dephosphorylated to Pant and then catabolized by vascular non-inflammatory enzymes (vanins) [31] that cleave the amide bond and generate cysteamine and Pan. A similar path is involved in the systemic degradation of CoA since isoforms of ENPPs and vanins are present both on the membrane of epithelial cells and in the interstitial spaces as soluble forms. Cysteamine has a free sulfhydryl group and may exert antioxidant activity. The upregulation of vanins-1 by oxidative stress [32] suggests an interesting link between CoA metabolism and oxidative stress, which adds up to the role of CoA as a substrate for protein coalation [33], a novel post-translational modification influenced by cellular redox status (see below). In the cells, the degradation of CoA proceeds in a similar way, with nudix (nucleoside diphosphate linked moiety X)-type motif hydrolases most probably involved in the conversion of CoA and acyl-CoA to P-Pant [1].

## 5. CoA Distribution and Usage in the Cell

Compartmentalization and inter-organellar fluxes represent an important mechanism to regulate CoA and CoA acyl-derivatives metabolism and functions. The highest levels of CoA are found in mitochondria (1–5 mM) and peroxisomes (0.7 mM), while lower amounts can be found in the cytosol (0.1–0.4 mM), the nucleus and the endoplasmic reticulum (ER) [1,6]. The inner membrane of mitochondria, peroxisomes and reticulum are impermeant to charged molecules, while nuclear pores and the outer mitochondrial membrane allow a free flux from and to the cytosol. Specific mitochondria and peroxisomes transporters for CoA/dephospho-CoA have been identified. The solute carrier family 25, member 42 (SLC25A42) catalyzes the import of CoA and/or dephospho-CoA into the mitochondrial matrix in exchange for adenine nucleotides [34]; interestingly, the fruit fly homologue appears to be specific for dephospho-CoA [35], implying a possible role for the molecule as a connector between mitochondrial and cytosolic pools of the cofactor. Another member of the same carrier family, SLC25A17, is localized in the peroxisomal membrane and carries CoA and other nucleotides inside the organelle with a counter-exchange mechanism [36]. Acetyl-CoA in the ER is required for protein and glycan acetylation. The transfer of CoA moieties into the lumen of the organelle is mediated by SLC33A1 that transports acetyl-CoA [37]. N-acetylation in the ER controls the proper folding and progression of several proteins to the Golgi apparatus, thus exerting a relevant quality check in the secretory pathway [38].

The levels of CoA and CoA acyl-derivatives differ significantly in animal tissues, with liver and cardiac muscle showing the highest concentration, followed by brown adipose tissue, kidney, and brain. CoA exerts a fundamental role in the intermediate metabolism acting as an activator of carboxylic acids in several biochemical transformations. The main CoA derivative is acetyl-CoA which is a metabolic intermediate in the production of energy from the catabolism of carbohydrates, lipids, protein and ethanol; it also provides the essential substrate for anabolic reactions producing fatty acids, cholesterol, bile acids, ketone bodies, heme, and acetylcholine among others [39]. In this context, CoA and CoA thioesters intervene also as fundamental regulators by modulating the activity of multiple enzymes; for instance, CoA and acetyl-CoA control the decarboxylation of pyruvate by allosterically inhibiting and activating pyruvate dehydrogenase kinase (PDK), respectively [40]. On the other hand, malonyl-CoA is the main inhibitor of carnitine palmitoyltransferase 1 (CPT1) [41], which catalyzes the condensation of acyl-CoA with L-carnitine to form acylcarnitine, then transferred into mitochondria to sustain beta-oxidation. These regulation activities ensure that energetic substrates are processed for ATP generation when cells are in energy-low conditions but used for anabolic reactions when the energy stores are replete. Another important role of CoA is the transfer of acyl groups for the post-translational modification (PTM) of proteins. Once again, acetyl-CoA is the most common substrate of such an activity, with the transfer of the moiety to the side chain of lysines in histones representing the first identified and most investigated process [42,43]. The acetylation of histones extends the connection between CoA level and the control of cellular functions to the modulation of gene expression. More recent studies showed that proteins other than histones undergo post-translational acetylation, including metabolic enzymes [44] and signalling molecules [45]. The transfer of an acetyl moiety to a protein can affect its function by changing the catalytic activity, the localization, the interaction with other molecules and its half-life. Hence, it is a key mechanism to directly control cellular processes, including energy pathways, cell growth and mitosis, autophagy and apoptosis [40]. If acetylation is one of the best characterized PTMs of proteins, CoA mediates also the covalent, post-translational transfer of other lipid moieties to proteins and chromatin for the dynamic control of their function. Several acyl-CoA metabolites, integral to major metabolic pathways, including succinyl-CoA, propionyl-CoA, and crotonyl-CoA can be covalently attached to the side chains of lysines of proteins in the mitochondria, ER, cytosol and nucleus [46]. Also, longer acyl-moieties can be substrates of this process. It is estimated that more than 100 and 1000 proteins can be modified by attachment of the myristoyl and palmitoyl residues, respectively. Each type of protein lipidation is mediated by specific acylating and deacylating enzymes, but, in many cases, spontaneous, non-enzymatic attachment of the acyl moiety to the amino group of lysines is documented. In most cases, the availability of acyl-CoA, and hence of CoA, influences the level of protein acylation, thus connecting their function with the nutritional and energetic status of the cell. Furthermore, the conjugation of acyl-CoA thioesters to aminoacids such as glycine and glutamine represents a relevant detoxification process. The large majority of the reactions involving CoA take place through an exchange of the organic acid moiety without net consumption of the molecule. Yet the total level of CoA in cells and tissues undergo substantial physiological changes and respond to stimuli such as hormones, nutritional status and pathological conditions. A conservative estimation performed by Yang et al. [47] calculates that the entire CoA content of the body undergoes an acetylation/deacetylation cycle twice every minute. Thus, it is very likely that changes in CoA level can quickly affect several of the above-mentioned processes, and hence cell activity and survival. Interestingly, CoA can also be a direct substrate for biochemical modifications. This is the case for two types of protein PTMs: the 4′-phophosphopantetheinylation and the coalation. P-Pant is an intermediate of de novo CoA biosynthesis and a moiety that can be attached to selected proteins with net consumption of CoA and release of adenosine 3′,5′ diphosphate [48]. The mitochondrial acyl carrier protein (mtACP) is the best characterized among the few proteins known to undergo this type of modification. It is an essential component of mitochondrial complexes involved in the electron transfer chain, iron-sulphur cluster assembly and lipoic acid metabolism [49]. The transfer of the P-Pant moiety from CoA is catalyzed by the holo-ACP synthase and is required for activation of mtACP [50]. Protein coalation identifies a process in which the sulfhydryl group of CoA forms a disulphide bond with cysteine residues of proteins [33]. This reversible PTM is favoured by oxidative or metabolic stress and affects the activity of metabolic enzymes and signalling molecules, the localization, stability, and interactions of target proteins. Several lines of evidence indicate that the extent of protein 4′-phophopantetheinylation and coalation correlates with the intracellular level of CoA, and recent data propose that defects in mtACP 4′-phosphopantetheinylation may represent a unifying pathogenic event in different neurodegenerative processes [51] (see below).

## 6. CoA Biosynthesis and Neurodegeneration

Given the large diffusion of Pan in nature, disorders due to reduced CoA concentrations are not common and generally related to defects in the biochemical processes controlling its homeostasis, including the biosynthesis, degradation and distribution [15,52]. There are data indicating a relevant connection between improper intracellular CoA handling and myopathies: mutations in the *PPCS* gene, coding for the enzyme catalyzing the second step in CoA biosynthesis (Figure 1), cause an autosomal recessive dilated cardiomyopathy [53], whereas defects in SLC25A42, the mitochondrial transporters of CoA, leads to variable clinical phenotype with myopathy [54,55]. Diabetes, cancers and infectious disease are other clinical conditions in which CoA availability seems to play a relevant role, but the strongest association is with rare, inherited neurodegenerative disorders (Table 1).

In 2001, Zhou and colleagues [56] described variations in the sequence of *PANK2* gene in cases of a movement disorder characterized by neural death and detection of high iron level in the brain. The disease was then identified as Pantothenate Kinase Associated Neurodegeneration (PKAN, OMIM # 234200) and the category of Neurodegeneration with Brain Iron Accumulation (NBIA) proposed to include pathologies with overlapping clinical features and the sign of iron deposition in the cerebral parenchyma, particularly in the basal ganglia [57]. The unexpected discovery spurred new interest in the study of CoA biosynthesis and metabolism [1] and was soon corroborated by the identification of defective COASY activity in other cases of NBIA, then named COASY Protein Associated Neurodegeneration (COPAN, OMIM # 615643) [23,58].

## 7. PKAN

PKAN is a rare (estimated prevalence of 1-2/1000000 [59]), heterogeneous disease, but represents the large majority of NBIA cases [60]; the typical form presents in childhood with dystonia, dysarthria, rigidity and retinal degeneration; the progression is fast and leads to an early death. Atypical forms are characterized by later onset, slower progression and more varied clinical features with parkinsonism, neuropsychiatric disorders and cognitive decline [61]. Several mutations have been described; many of them are expected to lead to a null allele with the complete loss of protein function; others appear to be associated with the presence of residual protein level and function [62,63,64] and even to be able to partially rescue pathological phenotypes when overexpressed in specific experimental models [65]. This could have some relevance for therapeutic strategies (see below). A recent analysis of PKAN cases described in the literature found a relevant genotype-phenotype correlation with early onset and rapid progression significantly associated with cases carrying two loss-of-function alleles [66]. Magnetic resonance imaging (MRI) is an essential step in the diagnostic path, with visualization of the “eye of the tiger” sign, a region of hyperintensity within the hypointense medial globus pallidus (GP) [67]; the neuropathological assessment of genetically confirmed cases documented the accumulation of iron together with astrogliosis, microgliosis, degenerative neurons, and axonal spheroids in the GP [68]. Interestingly, lesions with accumulation of apolipoprotein E and ubiquitin were found in the same brain area and interpreted as signs of hypoxic/ischemic injury [69].

PANK2 is involved in the biosynthesis of CoA and it is generally assumed that low level of cellular CoA is the main determinant of PKAN (Figure 2); definitive confirmation of this hypothesis is still lacking, since measurements of the CoA concentration in cells from patients or in different experimental models were inconclusive.

Analysis of distinct intracellular CoA pools could provide more precise information in this regard, as indicated by the recent work by Alvarez-Cordoba and colleagues [64], but further work with improvement of the analytical procedures is necessary to confirm this evidence. Extensive experimental modelling of the disease has led to the identification of distinct biological processes which are affected by PANK2 deficiency. Several studies highlighted the presence of severe perturbations of mitochondrial activity, both in terms of energy production and of anabolic processes; this was often associated with altered iron homeostasis, increased oxidative stress, and defective fatty acid metabolism [60,70] (Figure 2). The molecular connections and the origin of these pathological features are undefined yet; given the involvement of CoA and its thioester derivatives in a large number of cellular processes, it is likely that several fundamental reactions and pathways concur to establish them. Notably, the changes of protein PTMs depending on acyl-CoA and CoA itself could represent a relevant and unifying disease pathomechanism [51,71] (Figure 2).

Several invertebrate and vertebrate animal models have been generated to investigate PKAN pathological processes and possible therapeutic strategies. The presence of a single *PANK* gene (*fumble*) in *Drosophila Melanogaster* allowed to investigate the effects of a defective CoA biosynthesis in the absence of possible compensatory mechanisms by other isoforms of the enzyme. In contrast with many other experimental settings, the hypomorphic fly model showed reduced CoA concentration, together with altered lipid homeostasis, increased sensibility to oxidative stress and augmented DNA damage; vacuoles indicative of neurodegeneration were evident in the retina and brain and associated with loss of locomotor function and reduced life span [72]. The role of Pank2 in embryonal development was analysed also in *Danio rerio*. The downregulation of *pank2* expression in zebrafish altered the formation of the central nervous system [73]; embryos injected with a specific morpholino showed hydrocephalus and a significant reduction of neural markers in the brain, particularly in the basal telencephalon, which should contain regions homologous to the pallidum [74]. At the same time, severe perturbations of the main vessels and the vascular plexus in the tail were also frequent, suggesting the involvement of the enzyme in vasculature development; interestingly, the overexpression of mutant forms of the human PANK2 in zebrafish embryos induced similar vascular abnormalities [75] and silencing of the enzyme in human umbilical vein endothelial cells resulted in defective angiogenic properties [76]. This aspect has not been analysed in other experimental settings and awaits further investigation.

Even though there are some relevant differences between the human and murine PANK proteins in terms of structure and regulation, cellular distribution and tissue-specific expression [16,29], PKAN mouse models provided important clues about the biological consequences of defective CoA biosynthesis. *PANK2* knock-out (KO) mice did not show the clinical manifestation of PKAN, even after one year of age, but were characterized by reduced weight and fertility, due to a block in the spermatogenesis, and, in some strains, mild retinopathy. MRI and histological exams of the basal ganglia were negative [20], even though mitochondria in this region had altered morphology and showed reduced oxygen consumption when analysed in vitro [17]. When fed a Pan-deficient diet, mutant mice died quickly, while wild-type mice developed motor deficit and azoospermia among other signs [77]. A different *Pank2* KO model had normal brain level of CoA [22], while a drop was observed either when PANK1 and PANK2 were selectively deleted from the neurons [78] or when the degradation of CoA was enhanced by the neuronal overexpression of a nudix hydroxylase [79]. This suggests the presence of a significant compensatory activity among PANK isoforms in mice, particularly in the brain, which might not be the case in humans. Importantly, none of the models and conditions showed iron accumulation. The apparent limitation of these animals as bona fide models for PKAN has been recently reconsidered thanks to an in-depth analysis of the GP of KO mice [27]. The study revealed the presence in this section of the murine brain of a specific molecular and biochemical signature, which could recapitulate key features of PKAN. In particular, the expression of *Ppcs* and *Coasy* genes was reduced, and associated with signs of perturbed iron homeostasis, including increased iron level. The activity of pyruvate dehydrogenase (PDH) was increased while that of mitochondrial complex I was reduced. Finally, the amount of dopamine receptors transcripts was higher in the GP, while tyrosine hydroxylase protein level was decreased in the whole brain, indicating a possible origin for the movement disorder. The work has some limitations since CoA levels were not analysed, some aspects are in contrast with data in the existing literature and mice have no evident neurological symptoms; nonetheless, in combination with other data obtained in cellular and animal models [51], this analysis suggests the central role of CoA-dependent PTMs in the cascade of events underlying PKAN pathogenesis (see below).

Also, unicellular experimental settings have been largely exploited to investigate the effects of PANK2 deficiency. In contrast with most multicellular organisms, several unicellular models showed signs of perturbed iron homeostasis. This was evident in a yeast strain with deletion of the unique *pank* gene and its replacement with a mutant form [80], where iron levels were significantly increased, but also in mammalian cells exposed to PANK2-specific siRNA [81] and in fibroblasts from PKAN patients [82,83]. In mammalian cells the phenotype was generally characterized by increased labile iron pool (LIP), with higher level of transferrin receptor 1 (TfR1) and lower of ferritin, eventually associated with higher amount of ferroportin mRNA. Mitochondrial iron-dependent biosynthesis was defective, as indicated by the reduction of aconitase activity and heme content. Many of these features were confirmed in most studies investigating PKAN neurons differentiated from induced pluripotent stem cells (iPSC), and were associated with mitochondrial dysfunction, increased oxidative stress, and reduced glutathione [64,83,84,85]. The analysis of CoA and its thioester derivatives in these cellular models was less common and provided contradictory results [64,85]. Interestingly, increased expression of TfR1was found also in fibroblasts from patients affected by NBIA disorders other than PKAN [71]. The phenomenon was due to defective palmitoylation of the protein leading to impaired trafficking and recycling and was proposed as a possible link between CoA deficiency and iron accumulation. A recent study found calcium phosphate aggregates in the mitochondria of glutamatergic neurons from PKAN patients, together with signs of altered calcium homeostasis [86]. Calcification in the basal ganglia was previously described [87,88,89] in some PKAN patients and confirmed in about 50% of the cases analysed in this recent work [86], thus indicating a potential harmful association among iron, calcium and defective PANK2. Several informative hints about the biochemical processes perturbed by limited CoA availability came also from the analysis of *Drosophila* Schneider’s S2 cells. With the exception of the perturbation of iron-related parameters, several of the features described in mammalian cells were previously described in S2 cells with significant reduction of CoA content obtained by silencing for *pank*/*fumble* or treating with HoPan. The cells showed reduced growth, mitochondrial dysfunction and increased sensitivity to ROS [72,90], but also less efficient DNA repair mechanisms and perturbed organization of F-actin filament, with increased level of phospho-Twinstar (Cofilin), a regulator of actin homeostasis, and altered formation of lamellipodia [91]. The same authors found a reduced capacity to develop neurites in neuroblastoma cells silenced for PANK2. As expected on the basis of the multifaceted role of CoA in cell biology, the investigations performed in the above-mentioned experimental models documented a large variety of consequences induced by PANK2 loss of function and defective CoA biosynthesis. Nonetheless, a detailed description and comprehension of the molecular connections linking the initial defect with the varied cellular or organismal phenotypes are often missing or speculative. From this point of view, an important advancement came from a recent study in *Drosophila* cells [51], which investigated the connection between CoA availability and the post-translational 4′-phosphopantetheinylation of proteins. It showed that blockage of Pan phosphorylation hinders the transfer of the moiety to the mtACP. As mentioned above, this PTM is required for the activation of the carrier protein, which is a subunit of complex I in the electron transport chain and exerts several functions, including the synthesis of octanoic acid, a precursor of lipoic acid. Indeed, total protein lipoylation was reduced in S2 cells treated with HoPan, and among them the dihydrolipoamide acetyltransferase (PDH-E2), a subunit of the Pyruvate Dehydrogenase (PDH) complex. This was associated with a significant reduction of PDH activity, catalysing the transformation of pyruvate into acetyl-CoA, an essential step connecting the glycolytic pathway with the Krebs cycle (Figure 2). The results were largely confirmed in vivo (*Drosophila* model) and in mammalian cells and indicated a pathogenic path potentially linking neurodegenerative disorders with overlapping clinical phenotypes, namely PKAN and CoPAN, with defective CoA biosynthesis, MEPAN, a genetic disorder with altered production of lipoic acid, and a form of Leigh disease with mutations in PDH-E2. At the same time, the pathway has robust implications also for iron homeostasis and mitochondrial activity, since mtACP is involved in iron-sulphur cluster biogenesis [92] and complex I functioning and PDH activity is essential for the citric acid cycle. Proteins other than mtACP and PDH-E2 undergo 4′phosphopantetheinylation and lipoylation, so it is likely that other molecular mechanisms will be described to extend our comprehension of PKAN/CoPAN pathogenesis and possibly other disorders. The existence of a strong connection between the CoA/acetyl-CoA intracellular balance, the control of protein activity via lipidation and iron homeostasis is further suggested by the work of Petit et al., investigating the mechanisms of iron accumulation in Friedrich ataxia (FRDA) [93]. This is an inherited disorder due to reduced steady-state level of frataxin, a mitochondrial enzyme involved in iron-sulphur cluster biogenesis. The increase in mitochondrial and cytosolic iron levels in FRDA fibroblasts was linked to a dramatic fall in TfR1palmtoylation and defective recycling of the receptor. This feature, previously described in fibroblasts from other NBIA patients [71], was associated with decreased lipoylation and amount of PDH-E2 and reduced CoA levels. The requirement for iron-sulphur clusters for the activity of the lipoic acid synthase was speculated to be the trait d’union of the molecular path going from frataxin to iron accumulation, through reduction of PDH activity, altered control of acetyl-CoA/CoA ratio, and defective palmitoylation of TfR1 and other proteins. Treating fibroblasts with either CoA or dichloroacetate, which indirectly stimulate PDH activity, rescued the aberrant phenotype, indeed. Even though several aspects need to be better understood, these molecular mechanisms appear to connect and partially explain relevant features of PKAN pathology. The involvement of other biochemical pathways linked to CoA availability is also suggested by the metabolic profiling on plasma from PKAN patients, which revealed defects in fatty acids and cholesterol synthesis and bile acids conjugation [94].

## 8. CoPAN

The discovery that mutations in COASY [23,58,95], the last enzyme in CoA biosynthesis, lead to a disorder with clinical features partially overlapping with those of PKAN confirmed the central role of CoA-dependent biochemistry in selected forms of neurodegeneration. COPAN is a very rare, autosomic recessive disorder, characterized by early onset, mild oro-mandibular dystonia, dysarthria, spastic paraparesis, obsessive-compulsive behaviour, and cognitive impairment. The MRI usually shows hyperintensity and swelling of the caudate nucleus, putamen and thalamus; involvement of the GP may appear at later stages. The cases so far described carry the c.1495C > T (p.Arg499Cys) mutation either in homozygosity or in combination with other missense or nonsense mutations, all predicted to be pathogenic. Interestingly, the measurement of CoA level in fibroblasts from patients revealed no difference as compared to control cells, together with the presence of residual de novo CoA biosynthetic activity [23]. Mutations leading to a near complete loss of function of the enzyme result in prenatal onset of pontocerebellar hypoplasia [96], thus confirming the requirement for adequate level of the cofactor for the development of the central nervous system (CNS). Along this line of evidence, both deletion of the two enzymes (CAB4 and CAB5) exerting PPAT and DPCK activities in yeast [97] and effective downregulation of *coasy* expression in zebrafish embryos [98] were lethal, while expression of a mutant CAB5 or incomplete down-regulation led to milder phenotypes, with reduced CoA content. Di Meo et al. have recently developed a mouse model with selective deletion of the *Coasy* gene in neurons [99]. These mice showed a severe phenotype with arrest of growth, sensorimotor defects, and dystonia-like movements, early death, even in the absence of neuronal loss. Iron-related parameters were indicative of increased iron level that was confirmed by MS imaging of brain sections. Total CoA levels were unchanged, but mitochondrial oxygen consumption was reduced in the brain of KO mice. While supporting the existence of a strong relationship between CoA and iron homeostasis in the brain, the very short lifespan of these mice limits the investigation of the long-term, biochemical consequences of a defective neuronal production of CoA.

## 9. The Search of Therapeutic Approaches

The search for therapeutic strategies for PKAN and CoPAN moved from two main features of the disorders: the defect in a gene of CoA biosynthesis, potentially causing a reduction of CoA content in the cells or in selective subcellular compartments (mitochondria); and the accumulation of iron in the globi pallidi, the brain region most heavily affected by the neuropathology.

### 9.1. Correction of CoA Intracellular Content

As indicated before, most experimental settings modelling PKAN and CoPAN, and in particular the mammalian ones, did not document significant reduction of CoA content. At the same time, no data are available supporting alternative hypotheses, such as the existence of a moonlighting function for PANK2 or COASY, independent of CoA biosynthesis. Therefore, most investigations of potential therapeutic strategies for these human disorders aimed at reconstituting or increasing cellular CoA content (Figure 1). Most of the research dealt with PKAN and was grounded on different features of the disorder: it is due to a defect in the first step of CoA biosynthesis, multiple PANK isoforms can catalyse the same reaction, and some mutations in PANK2 appear to be compatible with the persistence of residual enzymatic activity. Besides the attempt to replenish cellular CoA content by providing the molecule itself, the other strategies tried to identify molecules capable of (a) boosting the residual activity of selective mutant forms of the enzyme, (b) bypassing the PANK2-dependent blockage in the biosynthetic pathway, and (c) stimulating other PANK isoforms to compensate for PANK2 deficiency. Different molecules were tested for their therapeutic potential, usually investigating their ability to prevent/abolish the phenotype of an experimental model and restore normal conditions (Table 1).

### 9.2. Boosting Residual PANK2 Expression/Activity

#### Pantothenate

High doses of Pan were shown to correct the aberrant features of PKAN fibroblasts and induced neurons carrying missense mutations of PANK2 and with some residual protein expression [64,100]. Adding vitamin B5 to the culture medium increased mRNA and protein levels of PANK2, restored normal mitochondrial CoA content, amount of mtACP, PDH, and complex I subunit ND1 among others, and prevented iron accumulation. The treatment was ineffective in fibroblasts carrying a frameshift mutation and with undetectable level of PANK2 protein. Interestingly, vitamin B5 corrected vascular abnormalities in zebrafish embryos overexpressing selective forms of human mutant *PANK2* genes but was ineffective in experimental settings based on downregulation of PANK2 expression [73,90,98]. Even though the available clinical experience appears to be negative, this approach deserves further investigation to establish its molecular basis and the connection with functional effects of the different missense mutations in PANK2 protein. The analysis of the residual PANK2 activity in patients’ erythrocytes could prove to be useful to this aim [101].

### 9.3. Bypassing the Blockage in the Biosynthetic Pathway

#### 9.3.1. Pantethine

Pantethine, the disulphide form of Pant, was one of the first molecules to be tested for its capacity to rescue PANK2-related phenotypes. The assumption is that it provides two Pant molecules for subsequent phosphorylation to P-Pant, thus overcoming the blockage in the biosynthetic chain. It was successfully applied in drosophila [90,102], zebrafish [73] and mouse models [103] but appeared to provide no benefit to patients in a recent open-label study [104]. This could be due to the activity of vanins in the serum, which cleave Pant to Pan and cysteamine.

#### 9.3.2. Fosmetpantotenate

Fosmetpantotenate is a membrane-permeable form of 4′-PPan, the product of PANK activity. It was shown to increase total CoA concentration and tubulin acetylation in neuroblastoma cells silenced for PANK2 and to permeate the blood brain barrier in orally dosed monkeys [105]. While pilot studies in few patients provided encouraging results [106], a phase 3 multicentre clinical trial failed to demonstrate clinical benefits in treated patients [107].

#### 9.3.3. Phosphopantetheine and Acetyl-4′-Phosphopantetheine

Supplementing exogenous CoA to the medium of *Drosophila* and mammalian cells with reduced PANK levels or treated with HoPan rescued their aberrant features, including reduced cell count and histone acetylation, and CoA deficiency [7,90]. It also prevented the premature death, the increase of ROS and the appearance of mitochondrial dysfunctions in PKAN iPSC-derived neurons, carrying alleles with premature stop codons [84]. When added to the food, CoA abolished the appearance of a compromised phenotype in mutant PANK or PPCDC flies but failed to do so in mutant COASY ones [7]. Interestingly, it largely restored the normal development of zebrafish embryos exposed to *coasy* specific morpholino. The mechanism of action in *Danio rerio* was not further investigated and could be due to the large permeability of embryos, while in cells, flies and mice it was shown that ENPPs degraded CoA to P-Pant. This molecule was stable in serum and in vivo, able to cross the cellular membrane and to restore CoA production when the blockage was due to defective PANK or PPCDC, but not to defective COASY. More recently, P-Pant in the diet corrected several molecular abnormalities detected in the GP of *Pank2* KO mice [27]. When the treatment was interrupted, the molecular perturbations reappeared within 7 days. A thio-acetylated form was shown to possess comparable rescue capabilities when assayed in PKAN fly or mouse models [108]. These encouraging results prompted the initiation of a clinical trial, which is actively recruiting (nbiacure.org/coaz-clinical-trial). Some issues about this potential therapeutic approach are still controversial [108,109]. The presence of a phosphate group should not be compatible with diffusion across the cell membrane and the stability of the molecule in culture medium or circulation should be reduced due to the activity of pantetheinases. Dephosphorylation eventually followed by degradation to pantothenate could intervene prior to utilization of the molecule and justify the rescue effect in the presence of residual PANK activity. Noteworthy, pantothenate was applied as a control in most experiments without any rescuing result.

### 9.4. Stimulating Other PANK Isoforms

Jackowsky’s group pursued an alternate approach and looked for molecules capable of boosting PANK activity. The idea was to compensate for PANK2 deficiency by the stimulation of the other PANK isoforms expressed in neurons. The optimization of the hits from a high-throughput screen of molecules [110] for their capacity to cross the blood brain barrier (BBB) led to identification of a set of lipophilic compounds, named pantazines, with high affinity for PANK enzymes [78]. The best hit, PZ-2891, acts as an allosteric activator of PANKs: it binds to one protomer of PANK dimer and locks it in an active conformation by inhibiting the binding of acetyl-CoA, which normally works as a negative feedback regulator. Treatment of human cells with the drug significantly increased CoA and acetyl-CoA content, in a PANK-dependent manner. A short treatment incremented CoA content in the liver and brain of mice and the drug itself was detected in the brain, thus confirming the crossing of the BBB. To prove the therapeutic potential of the drug, the authors developed a mouse model with neuronal deletion of both Pank1 and Pank2 genes. The animals had a short lifespan, reduced cerebral CoA amount, and locomotor defects. All these features improved significantly upon oral administration of PZ-2891. Further preclinical studies granted the start of a first-in human trial to evaluate the safety and tolerability of an improved pantazine (BP-671) in healthy volunteers (https://clinicaltrials.gov (accessed on 1 July 2021).

### 9.5. Iron Chelation

A common and distinctive feature of PKAN is the focal accumulation of iron in the GP. It is not clear yet whether it is an epiphenomenon, following neuronal death in the brain area that has a very high iron content physiologically or a mechanism elicited directly by the defect in PANK2 and contributing to the pathogenesis of the disorder. The recent data documenting a direct connection between defective PTM of proteins involved in iron recycling and iron sulphur cluster biogenesis [71,111] seem to support the latter hypothesis. Iron deposits have not been found in the brain of animal models of PKAN; more frequent is the documentation of increased iron content in cellular models [64,83]. Iron deposits are found in several neurodegenerative disorders. The metal can contribute to the development and progression of the pathology by exacerbating the generation of ROS, as documented by the increased presence of lipid, protein, and DNA adducts in the brain from patients (REFS). Furthermore, it has been associated with induction of neuroinflammation and protein aggregation. The availability of a potent iron-chelating agent capable of penetrating the BBB (deferiprone) and encouraging results obtained from sporadic treatments of patients with idiopathic NBIA [112,113] spurred the execution of the first clinical trials in a limited number of PKAN patients. Besides documenting safety and tolerability of the drug, they showed a decrease in globus pallidus iron content by MRI, but no or very limited clinical benefit in some patients [114,115,116]. The results from a larger, multicentre study were recently reported [117]. They confirmed the efficacy of the treatment at lowering iron content in the GP and evidenced a slower clinical progression, particularly in atypical PKAN patients. This supports the potential for iron redistribution from the brain as an approach to mitigate the ongoing damaging action of the metal and limit the progression of the disease [118].

## 10. Defects in CoA/Acetyl-CoA Intracellular Compartmentalization

### 10.1. CoA Transport to the Mitochondria

SLC25A42 is the best candidate transporter of CoA or dephospho-CoA from the cytosol to mitochondria [34,35]. A homozygous mutation potentially affecting the substrate-binding site of the protein was found in a 16-year-old child with myopathy and lactic acidosis [119]. Two other reports found the same mutation and an additional one causing an aberrant splicing of the mRNA in a larger set of patients, with variable clinical presentations, ranging from asymptomatic lactic acidosis to encephalomyopathy [54,55]. Interestingly, one of the patients showed symmetrical putamen atrophy combined with increased iron level in the GP and substantia nigra at the MRI. Modelling the loss of function of SLC42A25 in zebrafish embryos by microinjection of morpholinos resulted in bent tail, dorsal curvature, and delayed hatching from the chorion, indicative of motor defects. While the structure of the skeletal muscle appeared to be normal, mitochondria in the tissue showed swollen and fragmented inner membranes [119]. Measurement of total CoA content in fibroblasts from a single patient documented a 20% decrease as compared to control cells [54]. The transporter can carry molecules other than CoA in vitro, so interpretation of results cannot be definitive. Nonetheless, the data suggest that defective CoA/dephospho-CoA import into the mitochondria affects the organelle activity and homeostasis and results in clinical phenotypes of variable severity and possible perturbation of muscle and CNS development.

### 10.2. CoA Transport to Peroxisomes

SLC25A17, also known as PMP34, locates in the peroxisomes and can exchange CoA and other adenine nucleotides in reconstituted liposomes [36]. Zebrafish has two SLC25A17 homologues [120]. The downregulation of either of them led to overlapping developmental defects, with a larger yolk, retaining lipids even at 6 dpf, no inflation of the swim bladder, and impaired myelination of head and trunk nerves. Very long-chain fatty acids were increased and plasmalogens decreased, indicating peroxisomal dysfunction. The downregulation or overexpression of *slc25a17* in zebrafish embryos affected the expression levels of genes involved in lipid peroxisomal metabolism and in the development of endoderm-derived organs. Interestingly, the microinjection of CoA, but not of NAD^+^, rescued the phenotype observed in morphants. Together with in vitro transport assays, the data confirm the role of the carrier protein in peroxisomal CoA exchange. Surprisingly, knocking out *Pmp34* gene in Swiss Webster mice did not lead to a manifest phenotype [121]. KO mice had normal peroxisomal level of CoA, NAD^+^ and ATP and showed no change in very long chain fatty acids and ether phospholipids. Defects in the metabolism of phytanic/prystanic acids were evident only upon dietary administration of phytol. It seems unlikely that defects in SLC25A17/PMP34 could be associated with severe pathological conditions in humans, but further studies are necessary to clarify CoA metabolism in peroxisomes and its connection with the cytosolic and mitochondrial processes.

### 10.3. CoA Transport to the ER

Acetyl-CoA is required in the lumen of the ER for the acetylation of several resident or transiting proteins. Acetate can be attached to lysines in the polypeptide chain (Nε-acetylation) [122] or to the sialic acid in the carbohydrate moiety of glycoconjugates [123]. Acetyl-CoA enters the organelle thanks to the activity of a transporter, belonging to the SLC family, known as SLC33A1, ACATN1, or AT-1 [124,125,126]. The transporter plays a relevant role in CNS development, as suggested by the clinical manifestations found in patients carrying defective or duplicated alleles. Mutations leading to the absence or severe reduction of SLC33A1 protein cause the Huppke-Brendel syndrome, a rare autosomal-recessive disorder with developmental delay, hearing loss, congenital cataracts, low copper and ceruloplasmin. Hypomyelination and cerebellar hypoplasia can be detected by MRI [127,128]. Noteworthy, a specific loss of function, missense mutation found in the Chinese population (p.S113R) leads to an autosomal-dominant form of hereditary spastic paraplegia, named SPG42 [37]. On the contrary, gene duplication results in an autism-spectrum disorder with intellectual disability [129]. The genetic defects have been investigated in different experimental systems. Modelling the loss of function of the gene in zebrafish embryos by microinjection of a *slc33a1*-specific morpholino led to the appearance of a curly tail and defective axonal outgrowth from the spinal motor neurons [130]. Two knock-in (KI) mouse models carrying the p.S133R mutation were generated [131,132]. The *Slc33a1*^mut/mut^ homozygous mice died very early during embryogenesis. Heterozygous mice had slightly different phenotypes, probably depending on the strain. At one year of age, they showed variable motor impairment, eventually associated with a sensory deficit, but with no muscular involvement. Demyelination, neuronal loss and signs of neuroinflammation were detected in the spinal cord white matter or in the sciatic nerve. The p.S133R mutation affects the dimerization of the transporter in the ER membrane, resulting in a reduction of acetyl-CoA influx into the organelle and reduced acetylation of ER proteins. As previously shown in cell cultures [126,133], this leads to an aberrant induction of autophagy, via the defective acetylation of Atg9A. The overexpression of AT-1 in neurons of transgenic mice caused an autistic-like phenotype, with altered synaptic plasticity and increased dendritic spines [134]. Acetyl-CoA transport into the ER was increased and associated with a modified acetylome, a more efficient secretion of proteins and lower levels of cytosolic acetyl-CoA. The increased expression of SLC25A1, the transporter that translocates citrate from mitochondria to the cytosol, and ACLY, which converts mitochondria-derived citrate into acetyl-CoA, could represent mitochondrial compensating mechanisms. Altogether, these studies indicate a relevant role for the ER acetylation machinery in controlling the efficiency of the secretory pathway and the disposal of protein aggregates in the ER [135]. Changes in the balance of this process have profound and widespread proteomic and metabolic effects, which reflect the tight functional connection among different cellular organelles in the control of CoA/Acetyl-CoA homeostasis. This was further supported by an in-depth analysis of mice carrying the AT-1 mutant allele or overexpressing the wild-type one [136]. The two models showed opposite biochemical features: haploinsufficient mice had high cytosolic levels of acetyl-CoA, fatty acids and triglycerides, together with increased amounts of TCA cycle metabolites. Transgenic mice had lower cytosolic levels of acetyl-CoA and did not accumulate lipids in the liver, even when challenged with a lipogenic diet. TCA cycle metabolites were not increased, but hepatocytes had a significant expansion of the mitochondria network. The proteome was profoundly affected in both models, even though in a wider fashion in AT-1 overexpressing mice. As expected, pathways related to protein secretion and mitochondria activity/adaptation were affected, but changes extended also to different metabolic processes. Changes in the acetyl-proteome reflected the modification of cytosolic acetyl-CoA availability, showing hypoacetylation in the overexpression model and hyperacetylation in the haploinsufficiency one. The data confirm the involvement of AT-1 and the ER acetylation machinery in the control of acetyl-CoA cellular balance, at least in mice.

## 11. Conclusions

The knowledge about CoA has grown significantly since its first description by Fritz Lipmann [137]. It is involved in a large number of biochemical transformations, acting as an acyl-group carrier and a carbonyl-activating factor. The discovery that mutations in genes involved in its biosynthetic pathway, PANK2 and COASY, or in its compartmentalization, SLC25A42 and SLC33A1, are associated with rare, inherited neurodegenerative or neurodevelopmental disorders gave a great impetus to studies aimed at understanding the pathophysiology of the diseases, with the development of several cellular and animal experimental models. These studies provided new, important insight into the intricate network of biochemical processes involving the free cofactor or its thioesters and were instrumental for testing potential therapeutic strategies. Ongoing clinical trials for PKAN clearly testify the progress made by the field, but further work is still needed for a more complete understanding of CoA biochemistry. For example, a definitive confirmation that defects in PANK2 and COASY lead to a decrease of intracellular CoA pools is missing yet. Similarly, there is no explanation for the selective vulnerability of specific brain areas in PKAN and CoPAN and little is known about the molecular mechanisms connecting the different intracellular CoA pools and the cellular adaptive responses elicited by changes in each of them. Now we have a larger number of pieces of the intricate biochemical puzzle with CoA at its centre, but further basic research is necessary to complete and connect them, forming a comprehensive picture.

## Figures and Tables

**Figure 1 brainsci-11-01031-f001:**
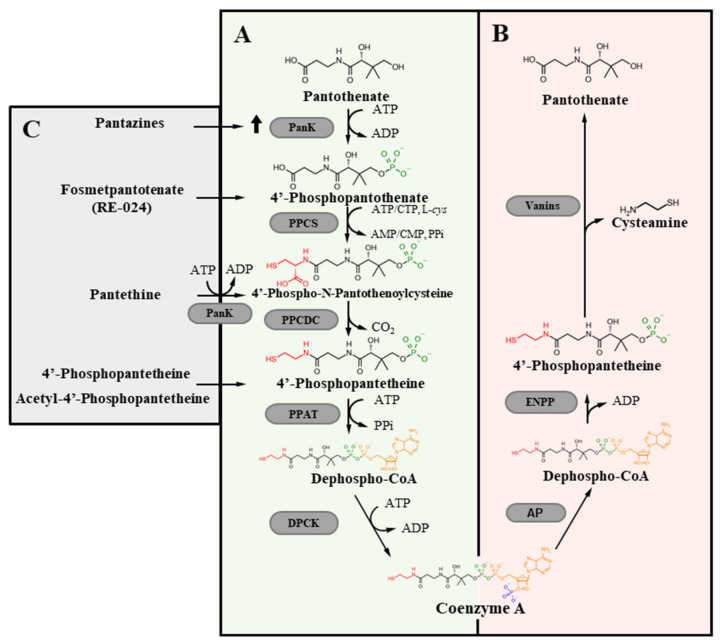
CoA homeostasis. Schematic description of the intracellular de novo CoA biosynthetic (**A**) and degradation pathways (**B**). Molecules shown to be able to correct defects due to a blockage in the first step of CoA biosynthesis (**C**). PANK, pantothenate kinase; Abbreviations: PPCS, phosphopantothenoylcysteine synthetase; PPCDC, phosphopantothenoylcysteine decarboxylase; PPAT, phosphopantetheine adenyltransferase; DPCK, dephospho-CoA kinase; ENPP, ectonucleotide pyrophosphatase/phosphodiesterase.

**Figure 2 brainsci-11-01031-f002:**
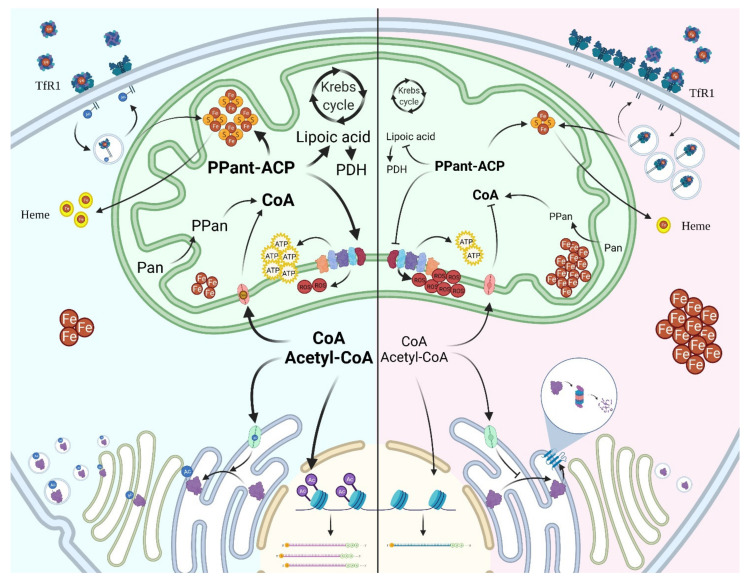
Schematic overview of the biochemical pathways involving CoA in normal (blue panel) or pathological conditions (red panel). The figure recapitulates some of the numerous intracellular processes requiring CoA or its thioester derivatives and found to be affected in disorders due to inborn errors of CoA biosynthesis or to altered compartmentalization. The important roles in the intermediate metabolism and mitochondrial energy production and in protein post-translational modifications (acetylation of histones and ER resident and transiting proteins, palmitoylation of TfR1, pantetheinylation of ACP) are summarized together with the effects observed in total or organellar CoA deficiency (iron accumulation, increased ROS production, defective energy production, reduced efficiency of protein secretion and ER degradation). Abbreviations: Pan, pantothenate, PPan, phosphopantothenate, PPAnt-ACP, phosphopantetheinyl-Acyl Carrier Protein, PDH, pyruvate dehydrogenase, TfR1, Transferrin Receptor 1.

**Table 1 brainsci-11-01031-t001:** Genes involved in CoA/Acetyl-CoA biosynthesis or transport and associated neurological disorders.

Gene (OMIM *)	Protein Function	Disorder (OMIM °)	Inheritance	Clinical Features	Molecules Investigated for Therapeutic Potential
PANK2 (606157)	PANK2 catalyzes the phosphorylation of pantothenate to 4-phosphopantothenate, first step of CoA biosynthesis.	Pantothenate kinase-associated neurodegeneration -PKAN (234200)	AR	Early (childhood) or late (early adulthood) onset. Dystonia, spasticity, parkinsonism, retinal degeneration, cognitive decline, neuropsychiatric disturbances.	Pantothenate; Pantethine; Fosmetpantotenate; 4′-Phosphopantetheine and Acetyl-4′-Phosphopantetheine; Coenzyme A; Pantazines; Artesunate; Deferiprone
COASY (609855)	COASY catalyzes the two final steps of CoA biosynthesis.	COASY Protein-Associated Neurodegeneration–CoPAN (615643)	AR	Early onset gait impairment and learning difficulties, with dystonia, and spasticity.	Coenzyme A
SLC25A42 (610823)	Mitochondrial CoA transporter	Recurrent metabolic crises with variable encephalomyopathic features and neurologic regression -MECREN (618416)	AR	Usually childhood onset with episodic lactic acidosis. Possible developmental regression of motor and cognitive skills.	-
SLC33A1 (603690)	Endoplasmic Reticulum acetyl-CoA transporter	Congenital cataracts, hearing loss, and neurodegeneration -CCHLNDH-uppke and Brendel syndrome (614482)	AR	Severe psychomotor retardation, congenital cataracts and hearing loss. More variable neurologic features. Low copper and ceruloplasmin serum levels.	-
		Spastic paraplegia 42 (612539)	AD - rs121909484 (p.S113R) in the Chinese population.	Spastic gait, increased lower limb tone, hyperreflexia, and weakness and atrophy of the lower limb muscles.	-
		Autism-spectrum disorder with intellectual disability	AD (gene duplication)	-	-

OMIM: online mendelian inheritance in man, https://www.omim.org (accessed on 5 July 2021); * Gene MIM; ° Phenotype MIM; AR autosomal recessive; AD autosomal dominant.

## Data Availability

Not applicable.

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
