# Peer review of "Coenzyme a Biochemistry: From Neurodevelopment to Neurodegeneration"

_brainsci, 2021, doi:10.3390/brainsci11081031_

Round 1
Reviewer 1 Report
This review from Mignani et al does a great job in analyzing and summarizing a rather complex area of biomedical research. The different CoA dependent pathways and their nodes of intersection are nicely described. The discussion is fair and comprehensive. The figures summarize the concepts well. Other than a few typos (ex; acetyl-CoA and not acetil-CoA in Fig 2), the manuscript is well written and I have no major modifications to suggest.
Author Response
We are grateful to the reviewer for the positive comments to our effort. we corrected a few typos in the revised manuscript, including the mistake in figure 2.
Reviewer 2 Report
Thank you for the very well written review. May be additional illustrations and tables can summarize the large amount of information presented.
Author Response
We are grateful to the reviewer for the positive comments to our effort. We added a graphical abstract and a table recapitulating the features of the genes involved in CoA homeostasis and associated with neurodegenerative and neurodevelopmental disorders.